# What Happens in the Gut during the Formation of Neonatal Jaundice—Underhand Manipulation of Gut Microbiota?

**DOI:** 10.3390/ijms25168582

**Published:** 2024-08-06

**Authors:** Hongfei Su, Shuran Yang, Shijing Chen, Xiaolin Chen, Mingzhang Guo, Longjiao Zhu, Wentao Xu, Huilin Liu

**Affiliations:** 1Key Laboratory of Geriatric Nutrition and Health, Ministry of Education, Beijing Technology and Business University, Beijing 100048, China; suhongfei@st.btbu.edu.cn (H.S.); chenshijing@st.btbu.edu.cn (S.C.); chenxiaolin@st.btbu.edu.cn (X.C.); liuhuilin@btbu.edu.cn (H.L.); 2NHC Key Laboratory of Food Safety Risk Assessment, Chinese Academy of Medical Science Research Unit, China National Center for Food Safety Risk Assessment, Beijing 100022, China; yangsr0605@163.com; 3Food Laboratory of Zhongyuan, Key Laboratory of Precision Nutrition and Food Quality, Department of Nutrition and Health, China Agricultural University, Beijing 100193, China; xuwentao@cau.edu.cn

**Keywords:** bilirubin, gut microbiota, Jaundice, neonates, *Clostridium*

## Abstract

Jaundice is a symptom of high blood bilirubin levels affecting about 80% of neonates. In neonates fed with breast milk, jaundice is particularly prevalent and severe, which is likely multifactorial. With the development of genomics and metagenomics, a deeper understanding of the neonatal gut microbiota has been achieved. We find there are accumulating evidence to indicate the importance of the gut microbiota in the mechanism of jaundice. In this paper, we present new comprehensive insight into the relationship between the microbiota and jaundice. In the new perspective, the gut is a crucial crossroad of bilirubin excretion, and bacteria colonizing the gut could play different roles in the excretion of bilirubin, including *Escherichia coli* as the main traffic jam causers, some *Clostridium* and *Bacteroides* strains as the traffic police, and most probiotic *Bifidobacterium* and *Lactobacillus* strains as bystanders with no effect or only a secondary indirect effect on the metabolism of bilirubin. This insight could explain why breast milk jaundice causes a longer duration of blood bilirubin and why most probiotics have limited effects on neonatal jaundice. With the encouragement of breastmilk feeding, our perspective could guide the development of new therapy methods to prevent this side effect of breastfeeding.

## 1. Introduction

Jaundice is a prevalent symptom in neonates with yellowish discoloration in the sclera and skin due to high blood bilirubin levels. In 80% of newborns, jaundice is evident [1,2]: in most cases, the blood bilirubin concentration breaks the “Visible Jaundice” line (about 85.5 μmol/L [3]) on the second or third day, reaches a peak of about 205.2 μmol/L [4,5] on the fourth or fifth day, and then subsides within ten days without causing complications to the neonates [6] (Figure 1), which has been termed “Physiologic Jaundice” [7]. However, when jaundice appears too early, increases too rapidly or too high (blood bilirubin >256.5 μmol/L) or lasts for an abnormally long time, or also progressively worsens or is combined with pathologic factors, it is referred to as “Pathological Jaundice” [8,9,10,11]. Pathologic jaundice, if not treated promptly and effectively, can progress further to acute bilirubin encephalopathy (acute phase manifestation of bilirubin toxicity occurring within 1 week of birth) or kernel jaundice (chronic, permanent clinical sequelae due to bilirubin toxicity). If the early treatment stage is missed and the condition further deteriorates, irreversible neurological damage will occur, seriously jeopardizing the health of the newborn [12].

The occurrence of jaundice is particularly related to the feeding mode of neonates. Compared with formula-fed neonates, breastfed neonates are much more likely to have jaundice [4,13,14,15]. Neonatal jaundice related to breast milk feeding is termed “breast milk jaundice”, which forms a double-peaks pattern based on the blood bilirubin concentration curve [16] (Figure 1) and persists for several weeks or months after birth [17]. Differently, the term “suboptimal intake hyperbilirubinemia” indicates the inadequate intake of breast milk by the neonates [18], which is associated with the extremely high peak of blood bilirubin concentration on the 2nd–4th day [6,7] (Figure 1). Many studies have focused on the mechanism of breast milk jaundice, with some metabolites or enzymes in human milk proposed to account for it, such as 3α20β-pregnanediol [19], free fatty acids [20], glucuronidase [10], and epidermal growth factor [21]; however, most of them could not be universally confirmed [15,22].

The relationship between the gut microbiota of neonates and jaundice has been proposed since the 1960s [23], but the previous view was incomplete and contradictory with many phenomena due to the limited knowledge of the neonatal gut microbiota. Traditionally, it was proposed that bacterial strains of *Escherichia coli* and *Clostridium* species, which could produce β-glucuronidase and enhance enterohepatic circulation in the gut [24], could increase the blood bilirubin level, while *Bifidobacterium* and *Lactobacillus* as probiotics could help neonates in the fading of jaundice [25]. However, this hypothesis is contradictory to the fact that breastfeeding neonates possess more *Bifidobacterium* in their gut but are more likely to be jaundiced. Thus, the relationship between the gut microbiota and neonatal jaundice was doubted.

With the development of high-throughput sequencing and its application in genomics and metagenomics, a deeper understanding of the development, composition, and function of the neonatal gut microbiota has been achieved. With this knowledge, there is new evidence to indicate more important roles of the gut microbiota in the mechanism of physiologic jaundice and breast milk jaundice. Besides, many factors that are related to physiologic jaundice, such as breastfeeding or formula feeding, term or preterm, and the use of antibiotics or not, are also the factors that influence the development of the neonatal gut microbiota (Figure 2). For example, a case-control study approach and high-throughput 16S rDNA gene sequencing were used to examine the composition of the intestinal flora in the feces of newborn infants, and it found that *Bacteroidetes* could be used as biomarkers for identifying pathologic jaundice [26]. Further, the occurrence of jaundice and the development of the neonatal gut microbiota are coordinated with respect to time. These findings inspire us to construct a new comprehensive perspective on the microbiota and physiological jaundice and try to explain breast milk jaundice from a microbiota view.

## 2. The Gut Is a Crucial Crossroads for the Route of Bilirubin Excretion

### 2.1. The Route of Bilirubin Excretion

In the neonate, bilirubin is derived from the breakdown of hemoglobin released from senescent red blood cells, and the bilirubin without conjugation (unconjugated bilirubin) is poorly soluble and transported in the plasma via binding to albumin until it is taken up by hepatocytes at the sinusoidal plasma membrane [27]. In hepatic cells, bilirubin is conjugated by the enzyme uridine diphosphate glucuronosyl transferase 1A1 (UGT 1A1) to increase its solubility in water [3]. The conjugated product of bilirubin is mainly bilirubin glucuronides, which are then secreted to the gut via the bile duct with the bile acids.

In the gut, the flux of conjugated bilirubin is divided into two branches. One part of conjugated bilirubin is converted to urobilinoids [28] and then excreted with the feces; the other part is hydrolyzed to form unconjugated bilirubin by β-glucuronidase and then reabsorbed in the small and large intestine [29] entering enterohepatic circulation. Most of the reabsorbed unconjugated bilirubin is delivered to the liver and conjugated again [30].

### 2.2. The Gut Is a Crucial Crossroad of Bilirubin Excretion

In the route of bilirubin, the maximum rate of excretion can be limited by the rate of enzymatically controlled bilirubin conjugation [31,32] and the carrier-mediated active transport system involved in the biliary excretion of bilirubin glucuronide [27]. However, these two procedures are one-way pathways for bilirubin excretion. The flux of bilirubin from plasma to hepatocytes is bidirectional, but normally this uptake process is not saturated [27]. The gut is the only area where enterohepatic circulation may occur. If bilirubin enters enterohepatic circulation and is transported to the liver, the reabsorbed bilirubin will accumulate in the hepatocytes, which makes the flux of bilirubin from plasma to hepatocytes restrained and the level of blood bilirubin increased. Otherwise, if bilirubin does not enter the enterohepatic circulation, the excretion of bilirubin will be unblocked (Figure 3).

This crossroad is crucial, as the importance of bilirubin enterohepatic circulation for physiologic jaundice has been widely proven [33]. The enterohepatic circulation of bilirubin was first confirmed using an isotope-labeled bilirubin in the early 1960s. Lester and Schmid used radiochemically pure bilirubin-C^14^ to study the enterohepatic circulation of bilirubin in rat models [30]. In their studies, the rate of recovered radioactivity from bile and feces ranged from about 1:10 to 1:1, which means that the amount of bilirubin entering enterohepatic circulation is considerable and variable in different individuals. Also, in rat models, Orlistat [34] and zinc sulfate [35] were used to inhibit the enterohepatic circulation of bilirubin, and the blood bilirubin levels were efficiently decreased.

In humans, Gilbertsen and colleagues confirmed the enterohepatic circulation of bilirubin in adult humans using crystalline bilirubin labeled with N15 [36]. The proportion of bilirubin entering enterohepatic circulation is difficult to measure. Nevertheless, through the inhibition of the enterohepatic circulation of bilirubin, the importance of enterohepatic circulation for the blood bilirubin level and physiologic jaundice has been proven. Ulstrom and Eisenklam used activated charcoal [31] and Poland and Odell [33] used agar to reduce the enterohepatic circulation of bilirubin via oral administration to neonates. Both of them found significantly lower blood bilirubin level in treated neonates than in the control neonates.

### 2.3. Microbiota Is the Traffic Police

The microbiota plays at least two roles in the gut crossroad of bilirubin excretion. On the one hand, the development of gut microbiota and their production of β-glucuronidase influence the occurrence of physiologic jaundice. Previous studies using isotope-labeled bilirubin in the Gunn rats model proved that only unconjugated bilirubin could be reabsorbed and retained in plasma, while conjugated bilirubin could virtually not [30]. Thus, β-glucuronidase with the catalytic activity of the hydrolysis of conjugated bilirubin to unconjugated bilirubin is crucial for the enterohepatic circulation of bilirubin. Gourley and his colleagues used L-aspartic acid and enzymatically hydrolyzed casein to inhibit β-glucuronidase in breastfeeding neonates, and the results showed increased fecal bilirubin excretion and less jaundice [37]. Among different sources of β-glucuronidase in the gut, bacterial β-glucuronidase is the most important in the enterohepatic circulation of bilirubin, which has been confirmed by research conducted with germ-free rats [38,39].

On the other hand, previous research has indicated that some specific strains of gut microbiota could convert the bilirubin to urobilinogen and then urobilinoids. Recently, *bilR*, a gene encoding bilirubin reductase, was identified via experimental screening and comparative genomics to reduce bilirubin to urobilinogen [40]. The reduction of bilirubin to urobilinoids by the intestinal microbial bacteria represents a natural detoxification mechanism, as the latter substances are believed to be nontoxic due to their increased polarity and to be much easier to be excreted [41].

Thus, the direction of bilirubin in the gut is affected by, firstly, the bacterial β-glucuronidase activity to hydrolyze conjugated bilirubin to unconjugated bilirubin, which promotes the enterohepatic circulation of bilirubin, and secondly, the bacterial conversion of bilirubin to urobilinoids, which inhibits the enterohepatic circulation of bilirubin. The pace of the above two reactions influences the rate of bilirubin elimination in physiologic jaundice.

## 3. The Roles of Different Neonatal Gut Microbial Taxa in the Excretion of Bilirubin

### 3.1. The Bacterial Taxa That Could Produce β-Glucuronidase and Hydrolyze Conjugated Bilirubin

In bacteria, β-glucuronidases are encoded mainly by GusB or uidA genes. Although the length of β-glucuronidases from different bacteria varies, all of these β-glucuronidases contain three motifs, Glyco_hydro_2_C (catalytic domain), Glyco_hydro_2_N (sugar-binding domain), and Glyco_hydro_2 (immunoglobulin-like beta-sandwich). According to the KEGG Orthology Database, the genes of β-glucuronidases (K01195) are present in the genomes of more than 340 bacterial strains. These strains are distributed mainly in the phyla Actinobacteria, Bacteroidetes, Firmicutes, and Proteobacteria. Some famous gut microbial species are involved, such as *Escherichia coli* and *Faecalibacterium prausnitzii*.

Using a structure-guided screening method, Pollet and colleagues proposed an atlas of β-glucuronidases in the human intestinal microbiome. They identified 3013 total and 279 unique microbiome-encoded GUS proteins clustered into six unique structural categories. Most of these proteins were assigned to the genera of Bacteroidetes and Firmicutes phyla [42]. Besides the GusB and uidA genes, Gloux and his colleagues used a functional metagenomic approach based on intestinal metagenomic libraries to screen the genes with strong β-glucuronidase activity. They identified an H11G11 gene family that had distant amino acid sequence homologies and an additional C-terminus domain compared with previously known β-glucuronidases [43].

Using traditional culture-based methods, β-glucuronidase activities have been confirmed in various bacterial strains. Some of these strains could be found in the neonatal gut microbiota.

***Escherichia coli.** Escherichia coli* is the most certain species that could hydrolyze conjugated bilirubin, and many strains belonging to this species have been confirmed to possess β-glucuronidase genes and activities. In vivo validation by Rob and Midtvedt showed that monocontamination with *Escherichia coli* in germ-free rats gave β-glucuronidase activities corresponding to those of the conventional rats [44].

***Enterococcus.*** Leung and colleagues found that only 4 of 53 strains of *Enterococcus* presented with β-glucuronidase activities, and the activities were very low [45]. There is no other indication of β-glucuronidase activity in *Enterococcus* strains.

***Bifidobacterium***. For the genus *Bifidobacterium*, although some strains of *Bifidobacterium longum* are labeled with β-glucuronidase genes in the KEGG database, a detailed inspection indicates that the genomes of these strains possess only the Glyco_hydro_2_C motif instead of the whole β-glucuronidase gene. The β-glucuronidase activity in *Bifidobacterium* seems to be limited to strains of *Bifidobacterium dentium* [46] and *Bifidobacterium scardovii* [47].

***Lactobacillus***. The β-glucuronidase activity has been detected in the *Lactobacillus rhamnosus* R strain [48] but has not been detected in the *Lactobacillus rhamnosus* GG strain [49]. Low β-glucuronidase activities were detected in the *Lactobacillus gasseri* F71 strain and *Lactobacillus parabuchneri* G46 strain [50]. No evidence has shown β-glucuronidase activity in *Lactobacillus acidophilus*, *Lactobacillus casei*, *Lactobacillus plantarum*, and *Lactobacillus reuteri* strains.

***Clostridium***. Many researchers clarified the high β-glucuronidase activities of *Clostridium*, which could be at least comparable to, and in some cases higher than, those of *Escherichia coli* [45]. Most strains of *Clostridium perfringens* [51,52,53] and some strains of *Clostridium sordellii* [52], *Clostridium clostridiiforme* [53], *Clostridium paraputrificume* [53], *Clostridium beijerinckii*, *Clostridium difficile*, *Clostridium limosum*, and *Clostridium ramosum* [51] were found to produce β-glucuronidase.

***Bacteroides.*** There is very little indication that strains of *Bacteroides* have β-glucuronidase genes and activities. Baranowski and colleagues found some strains of *Bacteroides fragilis* and *Bacteroides vulgatus* that presented with β-glucuronidase activities [52], while the results of research by Nakamura and colleagues showed that most strains of *Bacteroides* species show very low β-glucuronidase activities.

**Other bacterial strains.** β-glucuronidase activity has been characterized in *Ruminococcus gnavus* E1 [54], *Thermotoga maritima* MSB8 [55], and strains of *Peptostreptococcus* [56].

### 3.2. Bacterial Taxa That Could Metabolize Bilirubin

Libor Vitek and J. Donald Ostrow reviewed the chemical characteristics and metabolism of bilirubin [57]. According to this review, the bacterial metabolism of bilirubin was mainly reduced. Bilirubin is first converted to mesobilirubin via two-step reduction, then converted to urobilinogen via two-step reduction, and at last, converted to stercobilinogen via another two-step reduction. The urobilinogen and stercobilinogen could be oxidized to form urobilin and stercobilin, respectively. The enzymes that catalyze these reactions have not been clearly studied.

At the present stage, only several strains of the *Clostridium* genus and *Bacterioides fragilis* have shown high efficiency in reducing bilirubin. Gustafsson and colleagues found that compared to conventional rats, germ-free rats could excrete more bilirubin and no stercobilinogen. Germ-free animals contaminated with *Clostridium ramosum* strain G62 could produce stercobilinogen in the gut and contamination with the combination of *Clostridium ramosum* and *Escherichia coli* at the same time could increase the production of stercobilinogen [23]. Watson and colleagues confirmed the bilirubin-metabolism ability of *Clostridium ramosum* strain G62 [58]. Fahmy and colleagues studied the in vivo bilirubin metabolism ability of strains of *Clostridium, Bacterioides*, and other genera; they found that strains of *Bacteroides fragilis* and *Clostridium ramosum* showed the highest bilirubin-conversion efficiency, *Clostridium tertium* showed low bilirubin-conversion efficiency, and *Clostridium welchii* and *Clostridium sporogenes* showed no bilirubin-conversion ability [59]. Midtvedt confirmed the bilirubin-metabolism ability of *Clostridium ramosum*, and *Escherichia coli*, *Bacillus cereus*, and *Lactobacillus casei* could enhance the conversion efficiency of bilirubin to urobilins [60]. Vitek and colleagues found the strain of another *Clostridium* species, *Clostridium difficile*, that had bilirubin-conversion ability [41]. These numerous points are further supported by the work of Brantley et al. [40].

### 3.3. Three Groups of Gut Microbiota with Different Roles

According to the above analysis, the bacteria in the gut could be classified into three groups according to their roles in bilirubin metabolism (Figure 4).

**Traffic jam causers.** Bacteria with β-glucuronidase activity could enhance the reabsorption of bilirubin and cause a jam in bilirubin excretion. *Escherichia coli* is the most important species of traffic jam causers. Besides, *Clostridium* strains without bilirubin-conversion ability, a few *Lactobacillus* strains, and strains of *Ruminococcus gnavus* could be classified into this group.

**Traffic police.** Strains of *Clostridium perfringens*, *Clostridium difficile*, *Clostridium ramosum*, *Clostridium tertium*, and *Bacteroides fragilis* are the main traffic police on the crossroad of bilirubin excretion. Although these bacterial strains also possess the β-glucuronidase ability, the unconjugated bilirubin could be subsequently converted to soluble pigments and excreted with feces.

**Bystanders.** The bacteria with neither β-glucuronidase activity nor bilirubin-conversion ability could be called bystanders of bilirubin excretion. Most traditional probiotics belong to this group, including strains of *Bifidobacterium* species, *Lactobacillus plantarum*, *Lactobacillus acidophilus*, and *Lactobacillus casei*. Among them, *Bifidobacterium* is more specific, and although it has no direct effect on bilirubin excretion, it has been shown that the decrease in its abundance is positively correlated with a decrease in the expression of genes related to the galactose metabolism pathway, which reduces the upstream production of UDP-glucose, which leads to the lack of UDPGA and affects the conversion of bilirubin [61]. *Bifidobacterium* promotes defecation, lowers the intestinal pH, and inhibits β-glucuronidase activity [62,63], thereby inhibiting bilirubin enterohepatic recycling and favoring bilirubin metabolism to some extent.

## 4. The Development of the Neonatal Gut Microbiota and Jaundice

### 4.1. The Development of the Neonatal Gut Microbiota

Whether the fetal intestine is sterile remains a controversy [64,65]. Anyway, on the 1st to 3rd day of life, there is the first batch of colonizing bacteria in the gut of neonates [66]. Due to the aerobic environment of the neonatal intestine, aerobes and facultative anaerobes are more likely to be found among this pioneer gut microbiota [64] (Figure 5). The first colonization is distinctive in different populations, as it is significantly affected by environmental factors. The reported colonization list involves *Escherichia coli*, *Klebsiella pneumoniae*, *Enterobacter*, *Enterococcus faecalis*, *Enterococcus faecium*, and so on (Figure 5) [67,68,69,70,71,72,73]. *Escherichia coli* is the most prevalent bacterial species among neonates, and the content of *Escherichia coli* in the gut is relatively stable during the first three months [64]; it may persistently provide β-glucuronidase for the deconjugation of bilirubin glucuronides and promote enterohepatic circulation (Figure 6A). Thus, the blood bilirubin level increases gradually at 3–5 days of life (Figure 1).

With the gradual consumption of oxygen, anaerobes begin to colonize the neonatal intestine from the 3rd day to one week [74]. Traditional culture methods found that *Bifidobacterium*, *Bacteroides*, and *Clostridium* were three dominant genera of anaerobes in this period [67,68,75]. Bacteria with the confirmed ability of bilirubin conservation, such as *Bacteroides fragilis*, *Clostridium ramosum*, *Clostridium perfringens*, *Clostridium tertium*, and *Clostridium difficile*, could be present in the gut microbiota of one-week-old neonates, inhibiting the enterohepatic circulation of bilirubin and promoting the excretion of it (Figure 6B), which may be one of the reasons that physiological jaundice begins to subside from one week after birth (Figure 1). This implies that early gut microbiome intervention may be promising for the treatment of neonatal jaundice [76].

### 4.2. Neonatal Gut Microbiota, Breast Milk Jaundice, and Suboptimal Intake Hyperbilirubinemia

Many studies have revealed that human breast milk could increase the proportion of *Bifidobacterium* during the first month of neonates. It has been considered that the microbiota of breast-fed infants are dominated by bifidobacteria [77,78,79] and the proportion of facultative anaerobic bacteria, such as streptococci, staphylococci, enterococci, lactobacilli, and enterobacteria, is relatively low, while the gut microbiota of formula-fed neonates are more diverse and include bacterial groups, such as *Bacteroides*, *Clostridium*, and Enterobacteriaceae, with no obviously dominating species [64].

According to our analysis, *Bifidobacterium* and *Lactobacillus* genera are neutral or only indirectly effective for neonatal jaundice. Because, after the preceding discussion, for the most part, these strains do not have the bilirubin-reducing capacity and do not produce β-glucuronidase (and thus are unable to hydrolyze bound bilirubin directly), they appear to be unrelated to bilirubin metabolism. The really beneficial bacteria for jaundice are those that can metabolize bilirubin to urobilinoids. Most known strains with this ability belong to *Bacteroides* and *Clostridium* genera, for which colonization is inhibited by breastfeeding (Figure 6C). Thus, in breast milk jaundice, the bystanders are predominant and both the traffic jam causers and the traffic police are inhibited; thus, a relatively moderate amplitude, but a longer duration, of blood bilirubin increase could be found in breast milk jaundice neonates (Figure 1). The effect of breastfeeding on the composition of the neonatal gut microbiota could be influenced by many other factors, such as geography and the diet of mothers. As a result, some reports have not found differences among the types of feeding. The increased bifidobacteria proportion induced by the mother’s breast milk is not an inevitable event, which may be the reason why some breastfeeding neonates develop breast milk jaundice while others do not.

There are no studies on the relationship between breastfeeding times and the development of the neonatal gut microbiota. Of the many common features of gastrointestinal physiology, microbiology, genetics, and diet, the pig is a valuable model for human nutritional research [80]. And it has been shown that early weaning causes changes in the intestinal corticotropin-releasing factor (CRF) system, which can lead to intestinal barrier disorders and intestinal flora imbalance in pigs [81]. We could conjecture that the limited nutrition input would delay the colonization and development of the gut microbiota. Thus, on the 7th day of life, the facultative anaerobic bacteria would still be dominating the gut microbiota of neonates who receive insufficient breastfeeding. Thus, the suboptimal intake hyperbilirubinemia could be explained by the prolonged facultative anaerobic bacteria-dominated gut microbiota that hydrolyze more conjugated bilirubin, enhance the enterohepatic circulation, and increase the bilirubin level on day 3–7 of life (Figure 6D).

## 5. The Role of Probiotics in the Treatment of Jaundice

Commonly used probiotics mainly include strains from *Bifidobacterium*, *Lactobacillus*, *Streptococcus thermophilus*, and *Saccharomyces boulardii*. None of these strains have been reported to be able to convert bilirubin to urobilinogen and urobilinoids. Besides, most strains of these genera or species do not produce β-glucuronidase as well, except for some *Lactobacillus* strains. Thus, theoretically, these probiotic strains will not have direct effects on physiologic jaundice and the level of blood bilirubin. However, there have been several reports to propose probiotics for the management of neonatal hyperbilirubinemia [82]. To solve this contradiction, we reviewed the research on the effects of probiotics on physiologic jaundice. In most cases, probiotics of *Bifidobacterium*, *Lactobacillus*, and *Saccharomyces boulardii* strains seem to be ineffective in reducing the level of blood bilirubin, though some of these strains could have protective effects, such as anti-inflammatory [83], on jaundice neonates.

Two systematic reviews of randomized controlled trials of probiotic supplementation for the prevention or treatment of jaundice in neonates were published in 2017. One, by Chen and colleagues, showed that probiotic supplementation not only decreases the duration of phototherapy but also decreases the total blood bilirubin level [84], but this was mostly in the state of concomitant phototherapy. When probiotics were used as a single-strain preparation rather than in combination, i.e., without phototherapy, their therapeutic effect on jaundice was poor [85]. Secondly, the study by Deshmukh and colleagues showed that limited low-quality evidence indicates the reducing effect of the duration of phototherapy in jaundice neonates mediated by probiotic treatment [86]. Thus, there is not sufficient evidence to prove that *Bifidobacterium* and *Lactobacillus* probiotic strains are effective in reducing the level of blood bilirubin, which is coincident with our previous analysis of their ability to metabolize bilirubin. Most of the known strains possessing bilirubin-conversion ability, such as strains of *Bacteroides fragilis*, *Clostridium ramosum*, *Clostridium perfringens*, *Clostridium tertium*, and *Clostridium difficile*, are unsuitable to be used as probiotics, to screen new bilirubin-metabolizing strains from natural lactic acid bacteria, or to construct genetically modified microorganisms with this ability, which could be a promising approach to establish new treatment methodology for neonatal jaundice. In summary, the direct effect of common probiotics on physiologic jaundice is limited.

Phototherapy, the simplest treatment for neonatal jaundice, converts unconjugated bilirubin to a conjugated form, which can then be metabolized to urobilin analogs by the “traffic police” [87]. However, extended phototherapy is associated with potential dangers, including water and electrolyte imbalances, diarrhea, skin lesions, and bronze baby syndrome [88], and thus, it is often accompanied by adjuvants to shorten the duration of treatment, such as ursodeoxycholic acid [89]. In addition, there is much evidence that probiotics can be used in conjunction with phototherapy to treat neonatal jaundice. However, the mechanism of action of such combination therapy is not clear, and the effects of phototherapy and the intestinal flora appear to be complementary. On the one hand, probiotic supplementation has been shown to improve recovery from neonatal jaundice and reduce the side effects of phototherapy by modulating bacterial colonization and enhancing immunity [90]. On the other hand, research has shown beneficial changes in the gut microbiota structure of neonates with jaundice after phototherapy [91], such as a significant decrease in the abundance of *Escherichia coli*, which is accompanied by a decrease in the activity of β-glucuronidase [92], thereby favoring the excretion of bilirubin. Based on this, the probiotic strain *Bifidobacterium animalis subsp. lactis* CP-9 obtained by Tsai et al. through screening has excellent anti-*E. coli* activity, which can synergistically improve the efficiency of phototherapy [93]. Similarly, *Saccharomyces boulardii* may promote treatment outcomes and prevent the recurrence of jaundice by modulating the gut microbiota of neonates in combination with phototherapy [94]. Perhaps, synergistic treatment of neonatal jaundice can be further achieved by considering strains from the favorable alterations in gut flora after phototherapy when screening for new bilirubin-metabolizing strains, in conjunction with phototherapy.

## 6. Antibiotics, Preterm Neonates, and Jaundice

Blakery and colleagues studied the influence of parenteral penicillin and gentamicin on the gut microbiota of pre-term neonates using culture-based methods [95]. This result indicated that parenteral penicillin and gentamicin may decrease the amount and diversity of the gut microbiota, and species of *Clostridium* were more affected than *E. coli*. Similarly, Fouhy and colleagues studied the effects of parenteral penicillin and gentamicin via high-throughput sequencing and quantitative PCR methods, and the results showed that antibiotic-treated neonates had significantly higher proportions of genus members of Enterobacteriaceae and significantly lower proportions of *Bifidobacterium* and *Lactobacillus* than the untreated neonates [96]. In addition, antibiotic treatment resulted in reduced diversity of *Bacteroidetes* and/or delayed *Bacteroidetes* colonization, which persisted for at least three months. Due to the important role of members of *Bacteroidetes* in the development of a healthy, stable gut microbiota, antibiotic treatment may result in an increased risk of developing type 1 diabetes, asthma, and allergic diseases [97].

Arboleya and colleagues studied the preterm neonates whose mothers received ampicillin. They found the intrapartum antimicrobial prophylaxis affects the gut microbiota by increasing of Enterobacteriaceae family [98]. Greenwood and colleagues also studied the influence of ampicillin and gentamicin use in preterm neonates on the development of the gut microbiota. Their results revealed that infants receiving early antibiotics for 5–7 days had an increased relative abundance of Enterobacter and lower bacterial diversity in the second and third weeks of life [99].

Bonnemasion and colleagues reported the effects of amoxicillin combined with netilmicin (BI group) and amoxicillin, cefotaxime, and netilmicin (TRI group) on the colonization of bacteria in the gut. They found that BI Group samples were colonized with *Klebsiella oxytoca*, amoxicillin-resistant *Escherichia coli*, *Enterococcus faecium*, and coagulase-negative staphylococci. In the TRI group, biodiversity of the gut microbiota was low, with the rapid growth of staphyloccoci and the occurrence of *Candida* species [100].

The above reports indicated that, generally speaking, the commonly used antibiotics in neonates, such as penicillin plus gentamicin, ampicillin plus gentamicin, and amoxicillin plus netilmicin, could reduce the biodiversity of the neonatal gut microbiota. The lower biodiversity would result in the insufficiency of oxygen consumption, thus delaying the development of the anaerobic-dominated community. The longer period of the Enterobacteriaceae-dominated community may deconjugate more bilirubin glucuronides and increase the enterohepatic circulation of bilirubin.

## 7. Conclusions

In this review, we present a new perspective on the influence of the neonatal gut microbiota on the mechanism of physiologic jaundice. From this new perspective, the gut is a crucial crossroad of the bilirubin route, and bacteria colonizing the neonatal gut could play different roles in the excretion of bilirubin, including *Escherichia coli* as the main traffic jam causers, some *Clostridium* and *Bacteroides* strains as the traffic police, and most probiotic *Bifidobacterium* and *Lactobacillus* strains as bystanders with no direct effect on the metabolism of bilirubin. This perspective could explain why most probiotics have limited effects on neonatal jaundice. In breast milk jaundice, bystanders are predominant and both traffic jam causers and traffic police are inhibited; thus, a relatively moderate amplitude but a longer duration of blood bilirubin increases could be found in breast milk jaundice neonates. In suboptimal intake hyperbilirubinemia and preterm jaundice neonates, the delay of gut microbiota development causes an extended period of the traffic-jam-causer-dominant phase; thus, more bilirubin is reabsorbed and enters the enterohepatic circulation, and the blood bilirubin level increases severely on the 3rd–4th days. With the encouragement of breastmilk feeding, jaundice would be more of a risk for neonates, and our perspective could provide a theoretical basis and guide the development of new therapy methods to prevent this side effect of breastfeeding. For instance, when screening for new bilirubin-metabolizing strains or constructing transgenic microorganisms with this ability, strains with favorable alterations occurring after phototherapy are preferred. Concurrent synergistic treatment in combination with phototherapy may be a viable way to establish new treatments for neonatal jaundice.

## Figures and Tables

**Figure 1 ijms-25-08582-f001:**
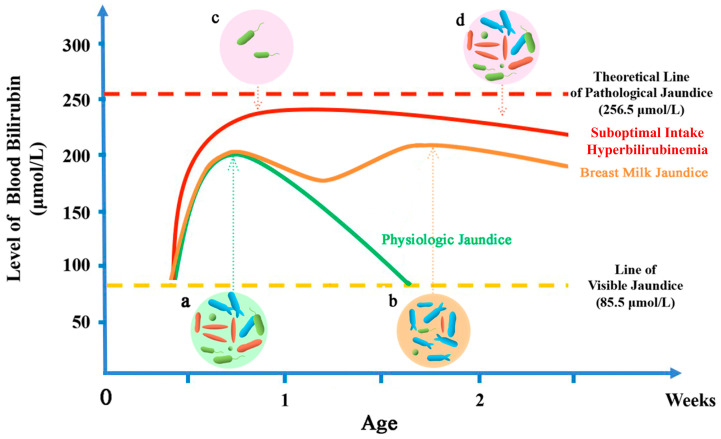
Blood bilirubin pattern of common physiological jaundice (green line), breast milk jaundice (orange line), and suboptimal intake hyperbilirubinemia (red line). (**a**) Anaerobic bacteria begin to dominate the gut microbiota at 4–5 days in the neonates with normal physiologic jaundice. (**b**) *Bifidobacterium* species begin to dominate the gut microbiota at 10–15 days in breastfed neonates. (**c**,**d**) The development of gut microbiota is delayed in the neonates with the inadequate intake of breast milk.

**Figure 2 ijms-25-08582-f002:**
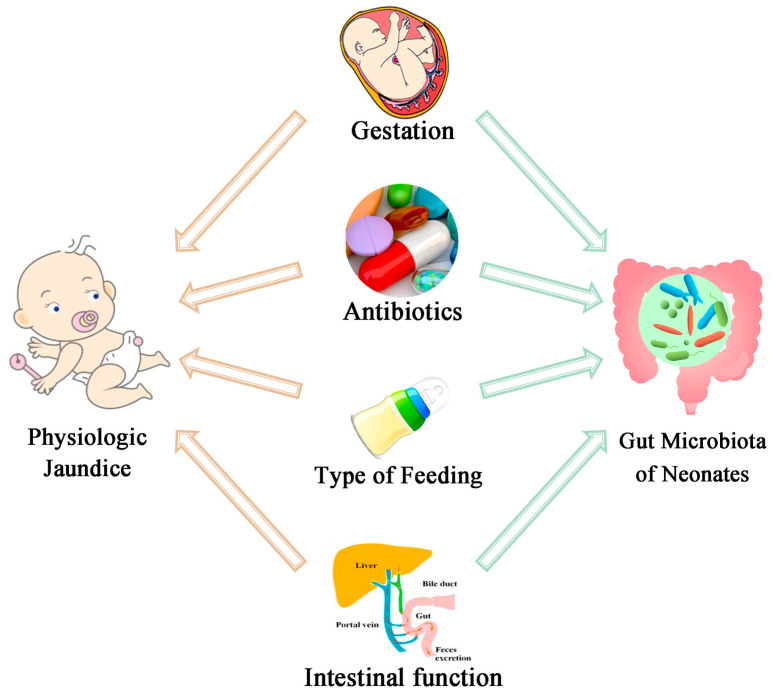
The factors that influence both the occurrence of physiologic jaundice and the development of the neonatal gut microbiota.

**Figure 3 ijms-25-08582-f003:**
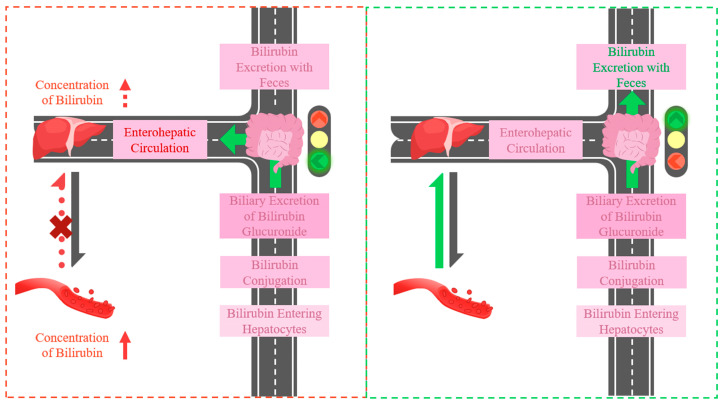
The route of bilirubin excretion. The gut is the key junction.

**Figure 4 ijms-25-08582-f004:**
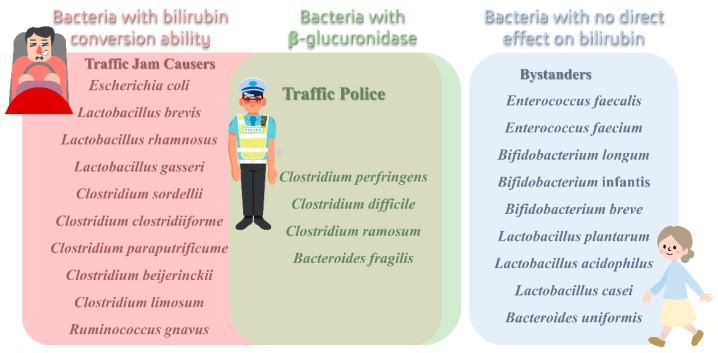
Different roles of gut microbiota. The red box is traffic jam causers, the green box is traffic police, and the blue box is bystanders.

**Figure 5 ijms-25-08582-f005:**
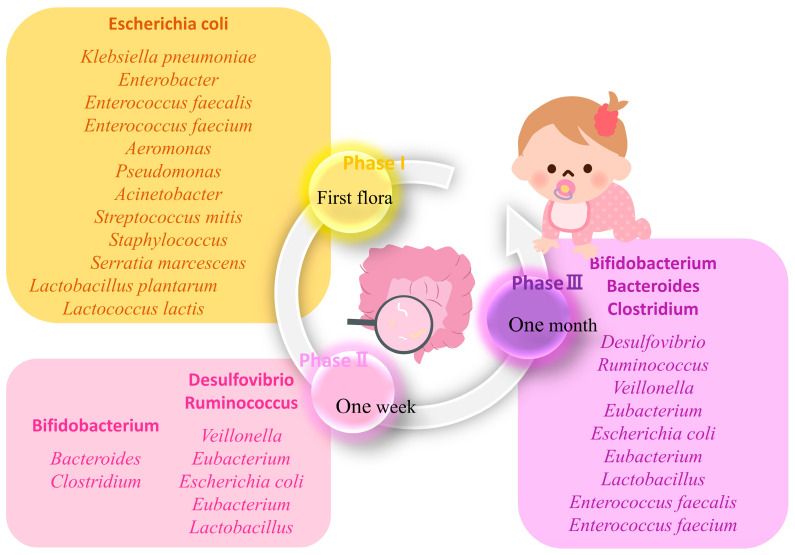
The development of the neonatal gut microbiota. The yellow box indicates stage 1 (first flora), the pink box indicates stage 2 (1 week), and the purple box indicates stage 3 (1 month).

**Figure 6 ijms-25-08582-f006:**
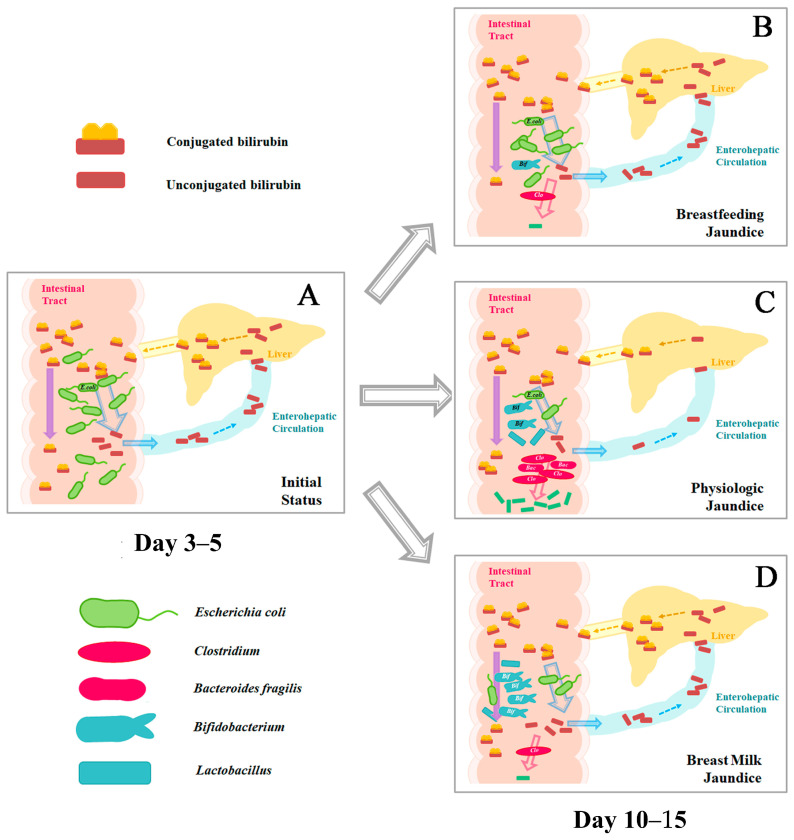
The development of neonatal gut microbiota impacts the occurrence of breast milk jaundice and the suboptimal intake of hyperbilirubinemia. (**A**) On days 3–5, the facultative anaerobes in the gut hydrolyze conjugated bilirubin and promote the enterohepatic circulation of unconjugated bilirubin. (**B**) The development of the gut microbiota is delayed in neonates with the inadequate intake of breast milk, which results in the continuation of enterohepatic circulation and occurrence of suboptimal intake hyperbilirubinemia. (**C**) On days 10–15, the anaerobes, including some *Clostridium* species and *Bacteroides* species, dominate the gut microbiota and convert the bilirubin to urobilinoids, which results in the disappearance of physiologic jaundice. (**D**) On days 10–15, *Bifidobacterium* species dominate the gut microbiota in breastfed neonates, the colonization of *Clostridium* and *Bacteroides* species is inhibited, and the excretion of bilirubin is blocked.

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
