# Peer review of "What Happens in the Gut during the Formation of Neonatal Jaundice—Underhand Manipulation of Gut Microbiota?"

_ijms, 2024, doi:10.3390/ijms25168582_

Round 1

Reviewer 1 Report

Comments and Suggestions for Authors

This study reviewed the role of gut bacteria (e.g., Clostridium, Bacteroides) in bilirubin excretion and metabolism, and developed the idea that bacteria (e.g., Escherichia coli) with bilirubin conversion ability as the traffic jam causers, bacteria (e.g., Clostridium, Bacteroides) with β-glucuronidase as the traffic police, and bacteria with no direct effect on bilirubin (e.g., Bifidobacterium, Lactobacillus) as as bystanders. Unfortunately, the present version of this work carries several prospective drawbacks.

1. The literature is too old, with only 3 references published after 2020. Thus, which cannot fully reflect the current research progress and is not timely. Please add articles published in the last three years.

2. Line 21, 391: Please revise the statement: “construct a theory” or “reconstruct a theory”. You just raised a new perspective. There is no confirmatory experiment, no conclusive evidence.

3. Line 31: Why use "Clostridium"? The role of Clostridium is consistent with Bacteroides, why emphasize Clostridium?

4. Line 390-406: It is recommended to add potential trends in future research, or to propose ideas for subsequent studies.

Comments on the Quality of English Language

Minor editing of English language required.

Author Response

Thank you very much for your valuable comments, which we have responded to individually, please see attached.

Reviewer 2 Report

Comments and Suggestions for Authors

In this article, the authors provided a new theory regarding the occurrence of neonatal jaundice and the development of gut microbiota in neonates.

Some suggestions are listed below.

1.      Line 33-41: The authors should provide a clear definition of pathologic jaundice. I think that a truly condition of pathologic jaundice is not discussed in this article.

2.      I suggest that the term “suboptimal intake-related hyperbilirubinemia” is to substitute “breastfeeding jaundice” in Figure 1, and in the main text as well.

3.      As shown in Figure 1, the bilirubin levels of the three kinds of jaundice all are lower than the theoretical line of pathological jaundice. Clinically, it is believed that physiologic jaundice and breast milk jaundice may not harm the baby. Therefore, the authors have to address the influence of neonatal jaundice on the neonatal health in the introduction.  

4.      Line 61-70: The authors have better provide appropriate references in the text.

5.      Line 289: “According to our analysis, Bifidobacterium and Lactobacillus genera are neutral or only indirectly effective to neonatal jaundice.” As being a new concept, the authors should provide more detailed information about their analysis.

6.      Line 302: “There are no studies on the relationship between breastfeeding times and the development of neonatal gut microbiota. We could conjecture that the limited nutrition input would delay the colonization and development of gut microbiota.” Is there any evidence to suggest the hypothesis?

7.      Line 334: “Two systematic reviews of randomized controlled trials of probiotic supplementation for prevention or treatment of jaundice in neonates were published in 2017.” Due to the controversial results, the author should provide more detailed information about the two reviews and try to explain why.

8.      Line 343-351: Do the authors have any suggestion about probiotics strains for neonatal jaundice?

9.      The paper should undergo some copy-editing. For example, the use of capital and small letters needs to be revised.

10.  The article would benefit from English language editing by a scientific editor.

11.  Minor suggestion: In Figure 3 and Figure 5, the words in white color were not comfortable to read.

Comments on the Quality of English Language

The article would benefit from English language editing by a scientific editor.

Author Response

(The authors gave the same response as above.)

Round 2

Reviewer 1 Report

Comments and Suggestions for Authors

This study reviewed and developed a perspective about the role of gut bacteria in bilirubin excretion and metabolism. The author has made some appropriate changes, but the “theory”(e.g., Lines 22-30, Line 90, Lines 440-453) still needs to be modified.

Comments on the Quality of English Language

Minor editing of English language required.

Author Response

Thank you for your valuable suggestion, please see attached for ours.

Reviewer 2 Report

Comments and Suggestions for Authors

I have no additional comments.

Comments on the Quality of English Language

I have no additional comments.

Author Response

Thank you for your earlier valuable comments on this article, which have contributed greatly to the quality of the text.